# Problematic Use of Video Games in Schooled Adolescents: The Role of Passion

**DOI:** 10.3390/bs14110992

**Published:** 2024-10-24

**Authors:** José María Faílde Garrido, María Dolores Dapía Conde, Manuel Isorna Folgar, Fátima Braña Rey

**Affiliations:** Faculty of Education and Social Work, University of Vigo, 32004 Ourense, Spain; jfailde@uvigo.es (J.M.F.G.); isorna.catoira@uvigo.gal (M.I.F.); fatimab@uvigo.gal (F.B.R.)

**Keywords:** video games, problematic use, addiction, adolescents, obsessive passion, harmonious passion

## Abstract

The present study aims to determine the predictive value of sociodemographic, academic, educational clinical psychological variables—ADHD diagnosis, emotional self-regulation, passion and emotional and behavioural symptoms—and modality of use of video games in adolescents who either make potentially problematic or severely problematic use of video games. This is a descriptive cross-sectional study in which 2.533 Galician students (region located in the northwest of Spain) aged between 11 and 20 years participated, who were selected through multistage random sampling. The mean age was 14.78 years (SD = 1.76). The analysis of the data seems to indicate that in the prediction of the problematic use of video games, different variables are combined, among which passion plays a capital role. In addition, the problematic use of video games is related to poor parental control, poor academic performance, use of addictive substances or to an earlier onset, as well as with the diagnosis of ADHD and with greater negative emotional and behavioural symptoms. The results of this study may be of interest for the design and implementation of preventive and corrective actions aimed at reducing the problematic or addictive use of video games.

## 1. Introduction

Video games have become a widespread leisure activity. According to the latest report by DFC Intelligence more than three billion people are regular consumers of video games, which represents almost 40% of the world’s population [1]. In Europe, according to the latest report by Europe’s video games industry [2], 51% of the population aged 6 to 64 play video games, with young people aged 11 to 14 doing so the most (with 84%), followed by the group aged 15 to 24 (with 75%), while in the group of 6 to 10 years also do so in a very prevalent way (with 73%). The data in this report indicate that as age increases, the use of video games decreases, with the age group of 45 to 64 years being the that which uses them the least (34%).

Spain follows a similar trend, as it can be seen in recent years. A recent study on digital entertainment among young people aged 15 to 29 shows that approximately 90% consume content related to video games [3]. This trend is like that observed in the ESTUDES survey, which reflects that 85.1% of students aged 14 to 18 have used video games during the last 12 months [4].

According to these data, we can affirm that video games are a playful practice that characterises the leisure of adolescents and young people. This extension of the use of video games not only refers to the fact that it is an activity mostly practised in this age group but also that it is carried out with great intensity given the time dedicated to it. Thus, according to the Spanish Observatory of Drugs and Addictions [4], almost half of the students aged 14 to 18 spend approximately two hours a day playing video games. Focusing on the group that makes potentially problematic use of video games, they dedicate 5 or more hours a day to gaming. In a similar way, Calderón and Gómez indicate that students dedicate 4.3 h per day to gaming [3].

For the vast majority of video game users, gaming constitutes a mode of entertainment without negative consequences [5]; however, the intensity in the involvement of young people—excessive use of video games—has caused significant concern and is becoming considered a behavioural addiction [6]. In this sense, the DSM-V of the American Psychiatric Association [7] has expanded and modified the category “substance-related disorders” to “substance use and addictive disorders”, in which it integrates the internet gaming disorder and defines it as “persistent and recurrent use of video games on the Internet, usually with other players, resulting in a clinical disorder or anguish, with five (or more) of the following symptoms, in a period of twelve months: worry, withdrawal syndrome, loss of control, loss of interest in previous hobbies, continued use despite having knowledge of their psychosocial problems, disappointment, avoidance and conflict in personal, work or academic environments” [7].

The prevalence of video game addiction is variable according to studies [8], being influenced by different factors and methodological issues, such as age, characteristics of the participants, country, game mode or instruments used. A recent meta-analysis places the global prevalence at 3.05% and 1.96%, being stricter in the selection of studies [9]. Similar values are confirmed in a meta-analytic study that quantifies the prevalence rates reported worldwide; it establishes the global prevalence grouped at 3.3% (8.5% in men and 3.5% in women), reducing to 2.4% in studies with representative samples, placing the adjusted prevalence at 1.4% through the application of stricter methodological calculation criteria [10].

In Spain, according to data from the Spanish Observatory of Drugs and Addictions [4], 7.1% of students aged 14 to 18 would present a possible disorder due to the use of video games (according to DSM-V criteria).

In recent years, research on video game addiction has increased, with interest in identifying what factors can predict this addiction. Different studies have associated sociodemographic, academic, parental control, playing manners, substance use or clinical variables (ADHD diagnosis, emotional self-regulation or emotional symptoms) with this behavioural addiction [11]. In line with a bibliographic review of existing scientific production until mid-2019, groups the different predictors of addiction to video games or problem gambling into the following: demographic, game characteristics, family influence, social influence, school environment, psychological and behavioural predictors and physiological indicators [11]. The conclusions of this study result in a majority profile characterised by being male, playing online, with low parental control and interest, little social interaction, negative school experience and low levels of emotional regulation.

The problems with the use of video games have been repeatedly associated in the scientific literature with the male gender, although it is not considered one of the most relevant factors. Thus, in a study using regression analysis conducted by Bernaldo de Quirós et al. [12], general mental health, gender and age only explain 1.3% of the total variance, while commitment to the game and cognitive distortions explain 59.3% of the variance of problematic video game use.

Other research suggests that passionate involvement in the use of video games can influence the development of problematic behaviours, although it depends on the type of passion that the player presents. Taking as a reference the dual model of passion (DMP) proposed by Vallerand et al. [13,14], harmonious passion (HP) and obsessive passion (OP) are distinguished. Regardless of the type of passion, it is defined by the authors as “a strong inclination towards an activity that people like, that seems important and in which they invest time and energy” [14], becoming, therefore, a meaningful, highly valued and self-defined activity that implies dedication and that causes satisfaction of basic psychological needs. Although at the level of functioning both types of passion follow the same scheme, HP is compatible with adaptive behaviours and individuals are able to maintain control in the execution of the activity that they are passionate about, not interfering with other responsibilities. On the contrary, OP causes individuals to lose control, causing a dependency that leads them to abandon other responsibilities. Some studies find that high levels of HP are associated with higher school engagement/success, while high OP is associated with low school engagement [15].

In a similar way, other studies have identified PH as protective and OP as a predictor of problematic behaviours and higher levels of abusive use of video games [16,17], associating low levels of satisfaction of needs in life with an OP for video games [18]. As a vicious circle, an intense OP towards gaming predicts and, in turn, is reinforced by a high frustration of needs [19], which in turn can generate OP and lead to problematic use of video games [20]. In short, playing video games with obsessive involvement can be interpreted as an escape route for a frustrated life that can favour problem gambling. In this sense, some authors consider the problematic use of video games as an emotional coping strategy [21].

The excessive use of video games is a complex and multifaceted phenomenon resulting from the interaction of various factors, including attention deficit hyperactivity disorder (ADHD), academic performance, parental control, gaming modality and substance use.

ADHD has been linked to a higher risk of developing video game addiction. Individuals with ADHD often struggle to control their impulses and maintain attention, which predisposes them to engage in activities that provide immediate gratification, such as video games. A recent study by Boer et al. [22] confirms that adolescents with ADHD are significantly more likely to engage in problematic video game use, using these games as a means to regulate their emotional state and avoid the frustration of academic or everyday tasks.

Additionally, academic performance is another factor that has shown a significant relationship with problematic video game use. Students who face difficulties in academics may use video games as a form of escape, which in turn exacerbates their academic performance. Recent research, such as that by Anjum et al. [23], has highlighted that excessive video game use is associated with a decline in academic achievement.

Parental control and supervision also play a crucial role in preventing problematic video game use. Krossbakken et al. [24] pointed out that adolescents with parents who implement screen time controls are less likely to develop video game addiction. However, when parents lack clear rules or are overly permissive, the risk of problematic use increases significantly.

The gaming modality is noted as another important factor related to video game use. In this regard, online video games, especially those with social components such as MMORPGs and battle royale games (e.g., Fortnite and League of Legends) have proven to be highly addictive. Montag et al. [25] indicated that features of these games, such as variable reward cycles, competitive social environments and the endless nature of gameplay, are factors that promote addiction. These games provide constant positive feedback and a sense of achievement, which can make it difficult for players to interrupt their sessions.

Finally, substance use has also been linked to problematic video game use. Individuals experiencing substance abuse tend to have a greater predisposition to other forms of addictive behaviour, such as excessive video game use. In a recent study [26] found that adolescents who reported excessive video game use also had a higher likelihood of engaging in performance-enhancing drug consumption.

In a literature review, we detected that few studies focused on analysing the joint influence of different variables with problematic use or addiction to video games. Thus, the objective of this study is to determine the predictive value of sociodemographic, academic, educational, clinical psychological variables—ADHD diagnosis, emotional self-regulation, passion and emotional and behavioural symptoms—and modality of video game use in adolescents who make potentially problematic use or severely problematic use of video games. To this end, a rigorous statistical methodology was used with a novel approach such as the Classification and Regression Tree (CART).

## 2. Methods

### 2.1. Participants

A total of 2533 participants took part in the study, after excluding 57 cases for exceeding 20 years or for not having completed any of the evaluation instruments. The mean age was 14.78 years (SD = 1.76), with a range between 11 and 20 years; all of the students were enrolled in secondary education (compulsory secondary education, high school and vocational and training cycles). Of these, 58.8% were male and 48.2% female.

For empirical contrast purposes, participants were classified into three groups based on the scores obtained in the Questionnaire of Experiences Related to Video Games [27]: (i) non-problematic use of video games (NPUVG); potentially problematic video game use (PPUVG); and severe or possibly addictive problematic use of video games (AUVG). A more detailed description of the characteristics of the participants can be seen in Table 1.

### 2.2. Instruments

For data collection, a set composed of the following instruments was administered:

Video Game Experiences Questionnaire (CERV) [27]. Instrument composed of 17 items, with a 4-point Likert response format (never/almost never, sometimes, quite often and almost always) that allows evaluating the problematic and abusive use of video games on any platform. It allows to obtain a total score and two subscales: evasion (8 items) and negative consequences (9 items). Likewise, cluster analyses offer a three-group solution based on the following cut-off points: no problems with the use of video games (scores between 17 and 25 points); potential problems (between 26 and 38 points); and severe problems (between 39 and 68 points). Cronbach’s alpha coefficients for the subscales are considered acceptable for the two subscales: negative consequences (α = 0.87) and evasion (α = 0.86). In this study, McDonald’s omega coefficients were 0.73 and 0.82 for each of the scales, respectively, and 0.89 for the global scale.

Emotional Repair Subscale of the Spanish version of the Trait Meta-Mood Scale (TMMS-249) [28]. It is composed of 8 items that evaluate metaknowledge for the regulation of emotions, with a 5-point Likert-type response format (from 1 strongly disagree to 5 strongly agree). The psychometric properties show adequate internal consistency (α = 0.86) for the emotional repair subscale; in this study, McDonald’s omega coefficient was also adequate (ω = 0.86).

Spanish version of the Strengths and Difficulties Questionnaire (SDQ) [29]. It is an instrument for clinical screening of mental disorders in childhood and adolescence, taking as a criterion the last 6 months. It is composed of 25 items, with a response format of three options (not true, a little true and absolutely true) grouped into 5 subscales (with 5 items each): emotional symptoms, behavioural problems, hyperactivity, relationship problems with peers and prosocial behaviour. A difficulty score can also be obtained, which is the sum of the previous subscales, except prosocial behaviour. The levels of reliability and validity for use in adolescents are adequate [29]. For this study, the McDonald’s omega coefficient obtained was 0.71 for the total difficulties scale and 0.72 for the prosocial behaviour scale.

Spanish version of the Escala da Pasión (adapted to video games) [30]. Passion is one of the elements of psychological processes present in various activities such as sports, leisure, work, interpersonal relationships or video games. This scale consists of 17 items, with seven answer options (strongly agree, fairly agree, agree, neither agree nor disagree, disagree, strongly disagree and strongly disagree). It allows us to obtain three dimensions: OP (6 items) and HP (6 items) and passion criteria (5 items). The internal consistency levels of the scale are adequate, with α = 0.81 for HP and α = 0.87 for OP. The values of the McDonald’s omega coefficient in the present study were suitable for both HP (ω = 0.73) and OP (ω = 0.75).

In addition to the above instruments, a questionnaire was administered, designed ad hoc, in which sociodemographic information was collected and on certain aspects related to the use of video games, parental control and the consumption of psychoactive substances.

### 2.3. Procedure

The selection of participants was carried out via multistage random sampling. First, from the list of centres of the Ministry of Culture, Education and University of the Xunta de Galicia (Government of Galicia, Spain), the number of students and centres in each province was determined and the number of centres (public and private/subsidised) per province in each of the educational levels (first stage) was selected. Secondly, the ratio of urban to rural centres by province (second stage) was established. Finally, we proceeded to select the class unit (third stage). Prior to the administration of the evaluation instruments to the participants, pertinent permission was requested from the parents through the direction of the educational centres.

The study was approved by the ethics committee of (masked for review purposes). This authorisation guarantees that the study complies with the ethical principles included in the Declaration of Helsinki for studies with human beings. All participants were informed of the purpose of the research and their written informed consent was obtained.

Data collection was carried out in groups in the presence of the researchers. The average time participants spent completing the instruments was 25 to 30 min.

### 2.4. Design

This is a descriptive study using cross-sectional surveys in which the independent variable was the type of player based on the cut-off scores of the Questionnaire of Experiences Related to Video Games (CERV) with three levels [27]: non-problematic use (NPUVG); potentially problematic use (PPUVG); and severe or possibly addictive problematic use (AUVG). The dependent variables were emotional self-regulation, symptomatology of emotional and behavioural problems, passion, sociodemographic characteristics, aspects related to the use of video games, parental control and consumption of psychoactive substances.

### 2.5. Data Analysis

After obtaining the data set and its subsequent digitisation, we proceeded to the treatment and analysis of the same.

First, the data were cleaned and coded for further analysis. Then, descriptive and exploratory analyses were carried out to determine the characteristics of the sample by calculating the measures of position, dispersion and shape, thus being able to establish the first results of homogeneity and presence of anomalous elements that could produce incoherent results. Next, a normality test was performed using the Kolmogorov–Smirnov statistic. Considering that the sample was not normally distributed, the Kruskal–Wallis test (non-parametric alternative to ANOVA) and the Mann–Whitney U test were used to perform post hoc contrasts (two to two). Likewise, the effect size was calculated using the Hedges *g* statistic.

To determine the levels of internal consistency of the instruments, we opted for McDonald’s omega coefficient, as it provides a more accurate estimate than Cronbach’s alpha. To do this, we followed the procedure indicated by [31].

In the final stage of the analysis, given that the dependent variable encompasses both quantitative and qualitative elements (once grouped), the decision was made to apply the classification and regression Tree (CART) statistical technique due to its ability to produce clear and understandable interpretations. The analysis included the following calculations:Error rate of the classification model: assessed to measure model accuracy.Classification rules for each observation: each observation enters the tree at the root and follows a descending path based on the values of its variables. At the terminal node, the classification percentage for each category of the dependent variable is determined.Variable importance in predicting the player type: the importance of each variable is evaluated based on its position in the tree, where variables located at higher levels carry more weight in the classification rules.Percentage probability of classifying a participant into a player category: the highest and lowest probabilities were analysed to identify the profiles of participants with the highest likelihood of belonging to a specific group or type of player.

The analysis process using classification trees was iterative. Based on an initial analysis of the influence of variables, various trees were developed using the complete set of independent variables, narrowing them down to those most representative for classification.

All analyses were performed using the statistical software SPSS 24.0 (IBM Corp., 2020).

## 3. Results

First, the results of the descriptive and influence analyses will be presented. Next, the classification and regression tree (CART) will be introduced.

### 3.1. Descriptive and Influence Analysis

The results of this section have been structured in three blocks: (1) Characteristics of the participants and patterns of use of video games; (2) consumption of psychoactive substances; and (3) psychological variables—passion and emotional and behavioural symptoms. For empirical contrast purposes, all variables were analysed based on the type of player (NPUVG, PPUVG and AUVG).

### 3.2. Participant Characteristics and Video Game Usage Patterns

As it can be seen in Table 1, a statistically significant association was detected between the variable type of player, with different variables that have been grouped into gender, academic performance, ADHD diagnosis, video game use and parental control, and that we describe as follows:

Gender (X^2^ = 403.08; *p* = 0.000)

The percentages of problematic and addictive use of video games are clearly higher in males, with 74.1% and 82.8, respectively. On the other hand, females mostly make non-problematic use of video games (67.2%).

Repeat course (X^2^ = 9.47; *p* = 0.050)

The participants of the AUVG group are those with the highest percentages of repeating a course (18.7%), while the PPUVG participants are the those who repeated two or more courses to a greater extent (7.1%). However, these data should be carefully observed, since the association between the repeats course variable and type of player is at the limit of statistical significance.

Number of failed subjects (X^2^ = 64.20; *p* = 0.000)

In line with the above, there is a clear statistically significant association between the number of failed subjects and the type of use made by the video game player. Thus, we identified that academic performance is worse in the PPUVG and AUVG groups compared to the NPUVG group.

ADHD diagnosis (X^2^ = 27.74; *p* = 0.000)

The data reflect a clear statistically significant association between the diagnosis of ADHD and the type of player. Thus, 7.2% of the PPUVG group have an ADHD diagnosis established by the national health system, a percentage that rises to 15.1% for the AUVG group.

Games for over 18 years (X^2^ = 222.02; *p* = 0.000)

The data show that those adolescents who deliberately opt for games for over 18 years (despite being minors) accumulate higher percentages of potentially problematic use or with severely problematic use.

Game mode (X^2^ = 213.26; *p* = 0.000)

The game mode (online versus offline) was significantly associated with the type of player. The online gambling modality accumulated a higher percentage of potentially problematic players (74.7%) and severely problematic players (78.9%).

Weekly play time (X^2^ = 495.34; *p* = 0.000)

We observed that the PPUVG and AUVG groups invest a greater number of hours of video game use. If we take a deeper look at the study of this variable, we observe the existence of statistically significant differences (Kruskal–Wallis test and post hoc contrasts, two by two, with the Mann–Whitney *U* statistic), depending on the type of player, in the number of weekly hours dedicated to the use of video games (*X*^2^ = 930.31; *p* = 0.000). The PPUVG (*p* = 0.000; *g*_+_ = 1.46) and AUVG (*p* = 0.000; *g*_+_ = −2.73) groups invested significantly more hours of play than the NPUVG group. Likewise, the AUVG group reported playing more weekly hours than the UPPVG, these differences being statistically significant (*p* = 0.000; *g*_+_ = −0.96).

Game parental controls (X^2^ = 146.13; *p* = 0.000)

Regarding the relationship between the type of player and parental control, we observed that the lack of parental control is related to higher percentages of potentially problematic (38.1%) or severe problematic (36.3%) use. Systematic parental control was also associated with moderate levels of potentially problematic (48.7%) and severely problematic (50.8%) use. On the other hand, partial (discretionary) parental control was associated with lower percentages of potentially problematic (13.3%) or severely problematic (11.5%) (12.9%) use.

Parental interest in play (X^2^ = 69.03; *p* = 0.000)

In a similar line to that observed in relation to the parental control variable, it is observed that the non-systematic or discretionary interest of parents in their children’s gambling was associated with lower rates of potentially problematic use (14.3%) or severely problematic use (13.7%).

### 3.3. Use of Psychoactive Substances

On the other hand, if we focus on the consumption of addictive substances in the last six months (see Table 2), we observe that alcohol consumption is more prevalent in the NPUVG typology (X^2^ = 7.06; *p* = 0.029). Regarding tobacco, the AUVG and NPUVG groups reported consuming alcohol more frequently (X^2^ = 8.67; *p* = 0.013), while the AUVG and PPUVG groups reported a higher consumption of energy drinks (X^2^ = 13.19; *p* = 0.001).

Statistically significant differences were also detected in the initiation of psychoactive substance use. Potentially problematic and addictive gambling are associated with earlier initiation of substance use (Table 2). Specifically, statistically significant differences were detected (Kruskal–Wallis test) between the type of player and the early onset of the following substances:

Alcohol (X^2^ = 15.16; *p* = 0.001)

Post hoc, two-to-two analyses with the Mann–Whitney U statistic show that players belonging to the NPUVG group have a significantly later onset of alcohol use. These differences were statistically significant with respect to the PPUVG (*p* = 0.038; *g*_+_ = −0.22 and AUVG (*p* = 0.000; *g*_+_ = −0.77) groups. Likewise, differences were detected between the ages of onset of PPUVG and AUVG (*p* = 0.013; *g*_+_ = −0.45), the latter being those with an earlier onset of alcohol.

Energy drinks (X^2^ = 21.84; *p* = 0.001)

NPUVG players start consuming energy drinks later, with a posteriori analysis showing that there are statistically significant differences only with respect to the AUVG group (*p* = 0.000; *g*_+_ = −0.37). Also, statistically significant differences were detected between the ages of onset of PPUVG and AUVG (*p* = 0.000; *g*_+_ = −0.45), the latter being those with an earlier onset in the consumption of energy drinks.

Psychotropic drugs (X^2^ = 14.77; *p* = 0.001)

In a similar line to what happened with alcohol, the post hoc contrasts reveal that NPUVG players have a significantly later start of the use of psychotropic drugs, reflecting the existence of statistically significant differences with respect to the PPUVG (*p* = 0.044; *g*_+_ = −0.21) and AUVG (*p* = 0.000; *g*_+_ = −0.67) groups. On the other hand, differences were detected between the PPUVG and AUVG groups (*p* = 0.009; *g*_+_ = −0.67); once again, it is the latter that starts earlier in the consumption of psychotropic drugs.

Psychological variables

On the other hand, as can be seen in Table 3, statistically significant differences were detected, depending on the variable type of player, in the levels of passion towards video games and in the emotional and behavioural symptoms.

In relation to the passion towards video games, statistically significant differences were detected depending on the variable type of player for both HP (X^2^ = 1539.98; *p* = 0.000) and for OP (X^2^ = 1563.16; *p* = 0.000). A posteriori analyses indicated statistically significant differences between the NPUV and PPUVG groups (*p* = 0.000; *g*_+_ = 2.55) and the AUVG group (*p* = 0.000; *g*_+_ = −6.37) in HP, the latter being the ones that referred to higher levels. Likewise, statistically significant differences were detected between the PPUVG and AUVG groups (*p* = 0.000; *g*_+_ = −2.15), the latter being the ones with the highest scores in HP. With regard to OP, the post hoc contrasts reflected the existence of statistically significant differences between the NPUVG and PPUVG groups (*p* = 0.000; *g*_+_ = 2.63) and the AUVG group (*p* = 0.000; *g*_+_ = 6.56). OP levels were also significantly higher in the AUVG group compared to PPUVG players (*p* = 0.000; *g*_+_ = 2.53).

Regarding emotional and behavioural symptoms (see Table 3), statistically significant differences were detected, depending on the type of player, in the following subscales of the Strengths and Difficulties Questionnaire (SDQ):

Emotional symptoms (X^2^ = 19.54; *p* = 0.000)

The AUVG and NPUVG typologies have the highest scores on this scale. Regarding the NPUVG group, their scores were significantly higher than those of the PPUVG (*p* = 0.000; *g*_+_ = −0.17). On the other hand, the AUVG group presented significantly higher scores than the PPUVG group (*p* = 0.005; *g*_+_ = 0.25).

Behavioural problems (X^2^ = 50.23; *p* = 0.000)

The AUVG group exhibited the highest scores on this scale, being statistically significant with respect to the NPUVG (*p* = 0.000; *g*_+_ = −0.47) and PPUVG (*p* = 0.005; *g*_+_ = 0.26) groups. Likewise, the PPUVG group had significantly higher scores than the NPUVG group (*p* = 0.000; *g*_+_ = 0.23).

Relationship problems (X^2^ = 13.36; *p* = 0.000)

Once again, the AUVG group obtained higher scores on this scale, being statistically significant with respect to the NPUVG group (*p* = 0.007; *g*_+_ = 0.24). The PPUVG group scored significantly higher than the NPUVG group (*p* = 0.004; *g*_+_ = 0.12).

Prosocial (X^2^ = 52.03; *p* = 0.001)

Unlike what happened in the previous scales, the NPUVG group obtained the highest scores in this dimension, with the differences with respect to the PPUVG (*p* = 0.000; *g*_+_ = −0.27) and AUVG (*p* = 0.000; *g*_+_ = −0.61) groups being statistically significant. Finally, the post hoc contrasts indicate the existence of statistically significant differences between the PPUVG and AUVG groups (*p* = 0.047; *g*_+_ = −0.30).

Classification and regression trees (CART)

After processing the data and completing the descriptive and influence analyses, the first analysis was conducted using classification trees. To evaluate the differentiated contribution of the variables to the model, they were incorporated into three sequential blocks. Due to space limitations, only the trees corresponding to step 3 are displayed.

In the first step, sociodemographic, academic, video game use and parental control variables were introduced as independent. At this stage, the model provided an overall percentage of classification that reached 70.2%; however, the model was not able to classify any case of the type of player belonging to the AUVG group.

In the second step, variables related to substance use, emotional self-regulation and emotional and behavioural symptomatology (the five dimensions of the Strengths and Difficulties Questionnaire) and ADHD diagnosis were introduced into the model. At this stage, the model remained at 70.2% of global classification and was not able to classify any case of the type of player belonging to the AUVG group.

In the last step, the variables related to passion towards video games (HP and OP) were introduced. The classification analysis, once all the variables were introduced (step 3), resulted in a model that significantly improved the percentage of correct classification, reaching 87.5% (See Figure 1). In this model, the risk estimate for a case is 0.125, which means that 12.5% of the cases were incorrectly classified (typical error = 0.007). In addition, the percentage of correct classification for each group was adequate: 89.6% for the NPUVG group; 90.4% for the PPUVG group; and 46.1% for the AUVG group.

It is important to bear in mind that, inside the error, the model errs in classifying only the adjacent classes, not in classifying any participant of the AUVG group in the category of NPUVG; similarly, no NPUVG was classified in the AUVG category.

In this model, it is observed that the variables related to the passion towards video games reached 87.5% (See Figure 1). In this model, the risk estimate for a case is 0.125, which means that 12.5% of the cases were incorrectly classified (typical error = 0.007). Video games are those that contribute relevant information. Therefore, the most important variables for the classification are OP and PH (see Figure 1). The main results obtained when analysing the tree are the following:The variable that matters most within the categorisation was the OP, presented at the first level.The most probable case of being a member of the NPUVG group can be extracted with a probability of 99.5%, defining a profile of adolescents who have low OP scores, located in quartile 1, and parental control, systematic or partial.The highest probability of belonging to the PPUVG group was set at 98.0%. It describes a type of player with OP scores located in quartile 3 and HP scores located in quartile 4.For the AUVG group, the most likely case was extracted with 50.4%, describing video game users with OP scores located in quartile 4, with HP scores located in quartile 4 and who play 6 or more hours.

## 4. Discussion

New technologies have caused changes in everyday behaviours, also modifying our leisure practices, particularly among younger generations. It is not possible to approach the leisure patterns of minors, adolescents and young people without taking into account digital leisure. A type of leisure that occupies a large part of the time and that presents different manifestations “allows to carry out traditional activities of the offline environment, but renewed or reformulated—listen to music, watch series or play video games, although, on the other hand, it allows to carry out activities that can only be carried out through the online medium—connect to virtual social networks or participate in virtual communities in a broad sense-…” [3]. In addition, such technologies allow their enjoyment at any time and anywhere. Internet access, therefore, opens a new field towards social, cooperative and competitive games, allowing users to interact with each other through so-called massively multiplayer online games [16].

In the scientific literature, there is no unanimous agreement on the terminology used to define players based on their mode of use of video games, using terms such as excessive use, problematic use, pathological use [32] or even addictive use [33]. In this study, we classified the players based on the cut-off scores of the CERV questionnaire [27], which establishes three types of players based on the following scores: (1) Non-problematic use; (2) potentially problematic use; and (3) severely problematic use.

The results of this study show that of the total number of participants who report playing video games, 46.0% make potentially problematic use, while 7% make severely problematic use, data that are in line with what has been reported in other studies [4].

In addition, consistent with what has been reported in the literature, problematic and addictive use of video games is more prevalent in males [12]. In this study, male adolescents are three times more prevalent in the PPUVG group than girls, while AUVG is five times more prevalent. This is something commonly reported in the scientific literature regarding addictive behaviours linked to substances, where use and abuse is more prevalent in males, with the exception of the use and abuse of hypnosedants [34,35]. Moreover, in terms of content preference, video games that promote competition, aggression and personal achievement tend to be more appealing to men. Additionally, men show a greater inclination toward action games and online multiplayer games, which tend to be more addictive due to their reward structures, intermittent reinforcements and competitive environments [36].

We have also observed that problematic and potentially addictive use is more prevalent among those participants diagnosed with ADHD [37]. This link can be explained by the difficulties individuals with ADHD have in regulating attention and impulses, as well as their tendency to seek rapid stimulation. In this regard, Nuyens et al. [38] indicated that adolescents with ADHD tend to prefer video games that provide immediate gratification and high levels of activation—such as action and multiplayer games—because they offer quick reward cycles and constant reinforcement. Additionally, Boer et al. [39] identified that problematic video game use in adolescents with ADHD may be influenced by the fact that video games provide a structured environment that these adolescents can control, unlike other areas of their lives that they often perceive as chaotic or difficult to manage. Video games provide a way to achieve success and control.

In addition, participants belonging to the PPUVG and AUVG groups have worse academic performance [40,41], lower parental control [42], a lack of concern for gambling recommendations based on their age and a greater predilection for online gambling [43].

Moreover, PPUVG and AUVG are associated with a higher prevalence of energy drink consumption [44]. Likewise, UPPVG and AUVG are associated with the earlier onset of certain addictive substances (alcohol, energy drinks and psychotropic drugs), which is consistent with studies that found a link between problematic video game use with consumption and earlier use of addictive substances [45,46]. Likewise, and in the same line, as has been reported in some studies, in terms of tobacco and video game use, it was found that other variables such as impulsivity may be common personality factors connecting both behaviors. Individuals with high impulsivity tend to be more likely to engage in both tobacco use and problematic video game use [47].

Regarding mental health, several studies have established the relationship between the problematic use of video games and its deterioration [48]. In this research, we found that PPUVG and AUVG present significantly higher scores than NPUVG, in most of the negative emotional and behavioural symptoms (emotional symptoms, behavioural problems, relationship problems) and lower scores in prosocial behaviour, data that are consistent with what was reported in studies such as García-Gil et al. [48].

Taking the type of player as a dependent variable, based on the criteria defined above, the CART analysis reflects that passion behaves as a factor of great relevance in the prediction and classification of the type of player. According to the dual model of passion [14], passion has a strong implication in identity, making exciting activities part of the player’s identity. PH predicts high levels of involvement in the use of video games; however, such individuals are able to reconcile the game with the optimal performance of other activities, allowing them to stop or moderate gaming if it were to have negative repercussions or cause serious interference with other important activities in their lives. On the contrary, OP causes serious interference with activities of daily living of such individuals, decreasing time to the detriment of these. Persistence is a consequence of this process, which becomes rigid and occurs, although the activity does not provide positive sensations and entails significant personal costs, such as the deterioration of relationships or less involvement in other tasks [13].

Although passion and addiction are different concepts, high levels of passion, especially OP, become compatible with potentially problematic or addictive behaviours. In our study, CART analyses indicated that AUVG was related to high levels of OP and PH, with the number of weekly hours spent playing a relevant role [49].

On the other hand, the model identified that the best ranking of players who make PPUVG corresponded to those who had moderate levels of OP and high levels of harmonious passion. The fact that both types of passion, OP and HP, are present in the prediction of potentially problematic use and, also, with severely problematic use, contradicts, in some way, the results of those studies that indicate that high levels of harmonious passion are linked to positive gaming behaviours and experiences (Fuster et al., 2014 [49]). However, this is consistent with what has been reported in recent studies that relate passion and the risk of addiction to video games. Thus, Szabo et al. [50] report high levels of OP and HP in people with exercise addiction, concluding that PH and OP could coexist as a single passion.

This study highlights that high levels of passion are related to problematic video game use [15]. Most likely, this is enhanced by marketing and the design of video games that make them highly attractive, with platforms, images and sounds of high quality. An aspect that can act on the ability to control the game, which, together with other personal and environmental factors, would facilitate the abusive and inappropriate use of these [24].

Although there is not much research related to the prevention of this problem [51], some initiatives, such as carrying out surveillance actions, prevention and treatment of internet addiction or legislating limited use in minors or controlling the time of use, could be effective in minimizing the negative impact of video game abuse [52].

In summary, the results of this study seem to indicate that in predicting problematic use of video games, different variables are combined, among which passion plays a capital role. In addition, the problematic use of video games is related to poor parental control, poor academic performance, ADHD diagnosis, time spent and game mode, consumption of addictive substances or with earlier initiations of consumption, as well as with greater negative emotional and behavioural symptoms [11,43,48]. The results of this study may be useful for the design and implementation of preventive measures, which should pay special attention to the passion variable, which when high (especially OP), in light of the results of this study, can be considered as a predictor variable of the risk of potentially problematic use of video games.

The strength of this study is in analysing the use of video games from a of multivariate analysis perspective, considering sociodemographic and academic factors, parental control, aspects related to the use of video games, substance use, passion and other clinical psychological variables. We also used a relatively large and representative sample of the population under study. However, it has several limitations that must be taken into consideration.

First, this study involved a cross-sectional design, which limits the ability to examine these issues from a more extended temporal perspective. Additionally, although the instruments used demonstrated good levels of internal consistency, this does not rule out the possibility that participants may have been influenced by social desirability bias. Therefore, future research should consider including additional measures to assess the impact of this bias. However, it is important to note that we made every effort to ensure that participants felt comfortable and free to express their responses, and that they perceived their anonymity to be fully guaranteed.

## Figures and Tables

**Figure 1 behavsci-14-00992-f001:**
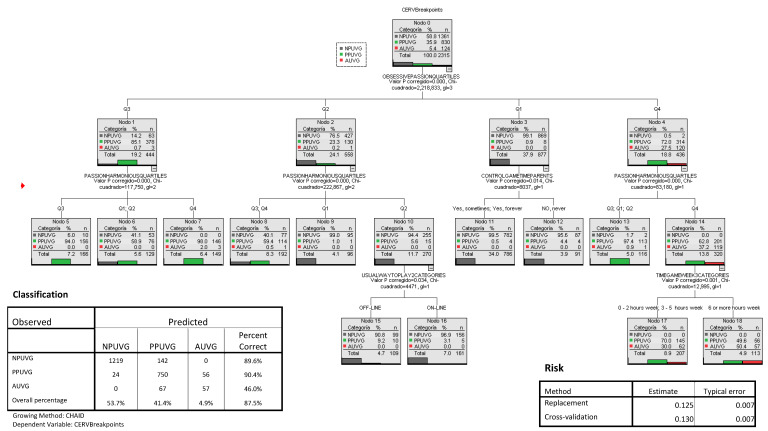
Classification and regression trees (CART) by type of video game player.

**Table 1 behavsci-14-00992-t001:** Characteristics of the participants (gender, academic performance, ADHD diagnosis, hours of video game use and parental control) depending on the type of players (chi square).

Variables	NPUVG N = 1357 F(%)	PPUVG N = 829 F(%)	AUVG N = 124 F(%)	X^2^	*p*
Gender					
Males	445 (38.6)	607 (74.1)	101 (82.8)	403.08	0.000
Females	911 (67.2)	212 (25.9)	21 (17.2)		
Repeat course					
No	1095 (80.8)	659 (79.5)	98 (79.7)	9.47	0.050
Yes. One Course	190 (14.0)	111 (13.4)	23 (18.7)		
Yes. 2 or more	70 (5.2)	59 (7.1)	2 (1.6)		
Number of failures					
4 or more	173 (12.9)	154 (18.8)	39 (32.0)	64.20	0.000
From 1 to 3	381 (28.4)	269 (32.9)	41 (33.6)		
Average performance	152 (11.3)	105 (12.8)	14 (11.5)		
Superior performance	634 (47.3)	290 (35.5)	28 (23.0)		
Diagnosis ADHD					
Yes	56 (4.2)	58 (7.2)	18 (15.1)	27.74	0.000
No	1273 (95.8)	745 (92.8)	101 (84.9)		
ADHD medication					
Yes	27 (47.4)	36 (62.1)	10 (58.8)	2.61	0.271
No	30 (52.6)	22 (37.9)	7 (41.2)		
Play over 18 games					
Yes	253 (21.3)	371 (45.4)	57 (47.5)	222.02	0.000
No	472 (39.7)	121 (14.8)	10 (8.3)		
I do not care	463 (39.0)	325 (39.8)	53 (44.2)		
Game mode					
Online	612 (45.0)	620 (74.7)	99 (78.9)	213.26	0.000
Offline	749 (55.0)	210 (25.3)	25 (20.2)		
Game time per week					
0 to 2 h per week	1193 (87.7)	458 (55.2)	35 (28.2)	495.34	0.000
3 to 5 h per week	136 (10.0%)	219 (26.4)	30 (24.8)		
6 or more hours per week	32 (2.4)	153 (18.4)	59 (47.6)		
Parents control game					
Yes. Always	895 (65.8)	404 (48.7)	63 (50.8)	146.13	0.000
Yes. sometimes	251 (18.4)	110 (13.3)	16 (12.9)		
Nope. Never	215 (15.8)	316 (38.1)	45 (36.3)		
Parental interest in game					
Yes. Always	727 (53.4)	311 (37.5)	52 (41.9)	69.03	0.000
Sometimes	208 (15.3)	119 (14.3)	17 (13.7)		
No. never	426 (31.3)	400 (48.2)	55 (44.4)		

Note: NPUVG: non-problematic use of video games; PPUVG: potentially problematic use of video games; AUVG: severe or possibly addictive problematic use of video games.

**Table 2 behavsci-14-00992-t002:** Substance use in the past 6 months (chi square) and ages of onset in substance use (Kruskal–Wallis test).

Have You Consumed in the Last 6 Months…?	NPUVG N = 1357 F(%)	PPUVG N = 829 F(%)	AUVG N = 124 F(%)	X^2^	*p*
Alcohol					
No	220 (28.6)	159 (35.8)	23 (34.3)	7.06	0.029
YES	549 (71.4)	285 (64.2)	44 (65.7)		
Tobacco					
No	124 (32.2)	89 (44.3)	8 (30.8)	8.67	0.013
YES	261 (66.8)	112 (55.7)	18 (69.2)		
Cannabis					
Yes	74 (40.0)	43 (43.9)	5 (38.5)	0.44	0.802
NO	111 (60.0)	55 (56.1)	8 (61.5)		
Energy drinks					
No	273 (34.3)	138 (25.6)	21 (24.1)	13.19	0.001
YES	523 (65.7)	401 (74.4)	66 (75.9)		
Psychotropic drugs					
No	80 (32.5)	30 (24.6)	6 (23.1)	3.01	0.222
YES	166 (67.5)	92 (75.4)	20 (76.9)		
**Age of onset of consumption of…**	**NPUVG** **Media (SD)** **N = 1361**	**PPUVG** **Media (SD)** **N = 830**	**AUVG** **Media (SD)** **N = 124**	**X^2^**	** *p* **
Alcohol	13.67 (2.07)	13.21 (2.60)	12.19 (3.23)	15.16	0.001
Energy drinks	12.80 (2.35)	12.58 (2.16)	11.60 (2.36)	21.84	0.001
Tobacco	14.20 (1.77)	13.85 (2.22)	13.96 (1.58)	3.38	0.184
Cannabis	14.99 (2.01)	14.89 (2.06)	14.30 (1.97)	3.15	0.207
Psychopharmaceuticals	14.25 (4.32)	13.39 (3.90)	11.88 (2.55)	14.77	0.001

Note: NPUVG: non-problematic use of video games; PPUVG: potentially problematic use of video games; AUVG: severe or possibly addictive problematic use of video games.

**Table 3 behavsci-14-00992-t003:** Passion, emotional self-regulation and emotional and behavioural symptoms depending on the type of player (Kruskal–Wallis test).

Variables	NPUVG Mean (SD) N = 1361	PPUVG Mean (SD) N = 830	AUVG Mean (SD) N = 124	X^2^	*p*
Harmonious passion	6.78 (1.04)	10.49 (1.95)	15.00 (2.84)	1539.98	0.000
Obsessive passion	7.21 (1.22)	11.22 (1.92)	16.35 (2.63)	1563.16	0.000
Emotional repair	24.37 (7.24)	24.92 (6.93)	25.03 (7.88)	4.19	0.123
Emotional symptoms	14.52 (4.82)	13.70 (4.72)	14.88 (4.75)	19.54	0.000
Behavioural problems	11.94 (2.86)	12.62 (2.93)	13.39 (3.34)	50.23	0.000
Relationship problems	15.42 (2.48)	15.72 (2.46)	16.04 (3.51)	13.36	0.001
Hyperactivity	16.08 (2.89)	16.31 (2.97)	16.40 (3.99)	5.17	0.076
Prosocial	21.53 (2.97)	20.68 (3.31)	19.62 (4.44)	52.03	0.000
Total score difficulties	57.98 (9.11)	58.37 (8.58)	60.73 (11.35)	16.42	0.000

Note: NPUVG: non-problematic use of video games; PPUVG: potentially problematic use of video games; AUVG: severe or possibly addictive problematic use of video games.

## Data Availability

The original contributions presented in the study are included in the article, further inquiries can be directed to the corresponding author.

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
