# Peer review of "Problematic Use of Video Games in Schooled Adolescents: The Role of Passion"

_behavsci, 2024, doi:10.3390/bs14110992_

Round 1
Reviewer 1 Report
Comments and Suggestions for Authors
The manuscript presented by the authors is of great interest at present, due to the high incidence of this problem in the adolescent population. The authors have carried out the study with a very large sample that includes 2533 adolescents, which is considered very significant. They have also distributed by groups according to degree of addiction, and have worked with numerous variables with which they have obtained many results with statistical significance, which have allowed very clear conclusions to be drawn.
The manuscript is very well structured and there are only small numerical errors on page 7, in the gender section and in the game parental controls section. The percentages mentioned by the authors do not coincide with those shown in Table 1.
In my opinion the manuscript is of great interest and should be in Behavioral Sciences, once the authors review the possible numerical errors corrected.
Author Response
Dear Reviewer,
Thank you very much for your comments and suggestions, which will undoubtedly contribute to improve the quality of this manuscript. Following your guidance, we have corrected the manuscript as you indicated, replacing the wrong data with the correct ones.
We have made these replacements and uploaded a new version of the manuscript. Additionally, we are attaching a version with track changes, where we have highlighted the modifications made in the manuscript text in red.
See section 3. RESULTS (pages 8 and 9)
Gender (x2=403.08; p=.000)
The percentages of problematic and addictive use of video games are clearly higher in males, with 74.1% and 82.8 respectively. On the other hand, women mostly make non-problematic use of video games (67.2%).
Game parental controls (X2=146.13; p=.000)
Regarding the relationship between the type of player and parental control, we observed that the lack of parental control is related to higher percentages of potentially problematic (49.0%) (38.1%) or severe problematic (56.1%) (36.3%) use. Systematic parental control was also associated with moderate levels of potentially problematic (38.3%) (48.7%) and severe problematic (36.9%) (50.8%) use. On the other hand, partial (discretionary) parental control was associated with lower percentages of potentially problematic (12.7%) (13.3%) or severe problematic (11.5%) (12.9%) use.
Reviewer 2 Report
Comments and Suggestions for Authors
The evaluated manuscript focuses on examining the factors that predict problematic or addictive video game use in Spanish adolescents, using the CART (Classification and Regression Trees) technique to analyze key variables such as ADHD diagnosis, emotional self-regulation, behavioral symptoms, parental control and passion. The subject matter of the article is highly pertinent, and the findings obtained have the potential to offer valuable insights for the formulation and implementation of prevention policies.
The manuscript provides a thorough and well-structured examination of problematic video game use among adolescents, and the approach to analyzing this issue is appropriate and relevant. However, one notable limitation is the lack of justification for the selection of most of the associated variables included in the study. While the role of sex and passion in problematic gaming is clearly explained and supported by previous research, other key variables such as ADHD, academic performance, parental control, game mode and substance use are not sufficiently justified in terms of their relevance to the phenomenon being studied. A more detailed discussion of why these specific variables were chosen, supported by relevant literature, would strengthen the rationale behind the research and enhance the overall clarity of the study.
The manuscript excels in its Method and Results sections. The methodology is robust, with a clear and thorough explanation of the multistage random sampling process and the use of the CART technique for data analysis. The large sample size and the rigorous approach to participant selection enhance the study's reliability. Additionally, the Results section is well-organized and presents the findings in a clear, concise manner, making it easy to follow the relationships between variables. The use of appropriate statistical techniques further strengthens the validity of the conclusions drawn. These sections significantly contribute to the overall quality of the manuscript. The only aspect I see as potentially improvable is to complement the reported reliability measures, such as Cronbach's Alpha, with McDonald's Omega. Given the limitations and assumptions inherent in using Cronbach's Alpha, incorporating McDonald's Omega would provide a more nuanced estimate of internal consistency, especially in cases where the data might not fully meet Alpha's assumptions. This addition would further strengthen the reliability assessment of the instruments used in the study.
In the Discussion section, the manuscript presents a new exposition of results by comparing them with findings from previous studies. However, this comparison lacks interpretation and does not provide possible explanations for the observed similarities or differences. For instance, there is no explanation regarding why males exhibit higher rates of problematic video game use or why video game addiction is more prevalent among adolescents with ADHD. In contrast, the discussion surrounding passion is well articulated and appropriately explained. Including a more analytical discussion of these issues, particularly for the variables that are not adequately addressed, would enhance the depth of the analysis and offer valuable insights into the implications of the results.
Author Response
Dear Reviewer,
Thank you very much for your comments and suggestions, which will undoubtedly contribute to improve the quality of this manuscript. Following your guidance, we have proceeded to correct the manuscript as you indicated, responding to all your comments and suggestions and incorporating the changes into the manuscript text. To this end, we have uploaded a new version of the manuscript. Additionally, we are attaching a version with track changes, where we have highlighted the modifications made in the manuscript text in red.
Comments and suggestions for the authors:
Q-1. El manuscrito proporciona un examen exhaustivo y bien estructurado del uso problemático de videojuegos entre los adolescentes, y el enfoque para analizar este tema es apropiado y relevante. Sin embargo, una limitación notable es la falta de justificación para la selección de la mayoría de las variables asociadas incluidas en el estudio. Si bien el papel del sexo y la pasión en los juegos problemáticos está claramente explicado y respaldado por investigaciones anteriores, otras variables clave como el TDAH, el rendimiento académico, el control parental, el modo de juego y el uso de sustancias no están suficientemente justificadas en términos de su relevancia para el fenómeno que se estudia. Un análisis más detallado de por qué se eligieron estas variables específicas, respaldado por la bibliografía pertinente, fortalecería la justificación de la investigación y mejoraría la claridad general del estudio.
Response Q-1:
We completely agree with your comment; therefore, we have incorporated the following text in Section 1. INTRODUCTION (See pages 3, lines 117-149).
The excessive use of video games is a complex and multifaceted phenomenon resulting from the interaction of various factors, including attention deficit hyperactivity disorder (ADHD), academic performance, parental control, gaming modality, and substance use.
ADHD has been linked to a higher risk of developing video game addiction. Individuals with ADHD often struggle to control their impulses and maintain attention, which predisposes them to engage in activities that provide immediate gratification, such as video games. A recent study by Boer et al. (2019) confirms that adolescents with ADHD are significantly more likely to engage in problematic video game use, using these games as a means to regulate their emotional state and avoid the frustration of academic or everyday tasks.
Additionally, academic performance is another factor that has shown a significant relationship with problematic video game use. Students who face difficulties in academics may use video games as a form of escape, which in turn exacerbates their academic performance. Recent research, such as that by Anjum et al. (2024), has highlighted that excessive video game use is associated with a decline in academic achievement.
Parental control and supervision also play a crucial role in preventing problematic video game use. Krossbakken et al. (2018) have pointed out that adolescents with parents who implement screen time controls are less likely to develop video game addiction. However, when parents lack clear rules or are overly permissive, the risk of problematic use increases significantly.
The gaming modality is noted as another important factor related to video game use. In this regard, online video games, especially those with social components such as MMORPGs and battle royale games (e.g., Fortnite and League of Legends), have proven to be highly addictive. Montag et al. (2019) indicated that features of these games, such as variable reward cycles, competitive social environments, and the endless nature of gameplay, are factors that promote addiction. These games provide constant positive feedback and a sense of achievement, which can make it difficult for players to interrupt their sessions.
Finally, substance use has also been linked to problematic video game use. Individuals experiencing substance abuse tend to have a greater predisposition to other forms of addictive behavior, such as excessive video game use. In a recent study, Ip et al. (2021) found that adolescents who reported excessive video game use also had a higher likelihood of engaging in performance-enhancing drug consumption.
References:
Anjum, R., Nodi, N. H., Das, P. R., Roknuzzaman, A. S. M., Sarker, R., & Islam, M. R. (2024). Exploring the association between online gaming addiction and academic performance among the school-going adolescents in Bangladesh: A cross-sectional study. Health Science Reports, 7(9), e70043. https://doi.org/10.1002/hsr2.70043
Boer, M., van den Eijnden, R. J., Boniel-Nissim, M., Wong, S. L., Mathews, C., Morrell, H., & Molle, J. (2019). Video game addiction, ADHD symptomatology, and video game reinforcement. The American Journal of Drug and Alcohol Abuse, 45(1), 67–76. https://doi.org/10.1080/00952990.2018.1472269
Ip, E. J., Urbano, E. P. T., Jacobs, R. J., Lau, W. B., Clauson, K. A., Torn, R. A., Palisoc, A. J. L., & Barnett, M. J. (2021). The video gamer 500: Performance-enhancing drug use and Internet Gaming Disorder among adult video gamers. Computers in Human Behavior, 123, 106890. https://doi.org/10.1016/j.chb.021.106890
Krossbakken, E., Torsheim, T., Mentzoni, R. A., King, D. L., Bjorvatn, B., Lorvik, I. M., & Pallesen, S. (2018). The effectiveness of a parental guide for prevention of problematic video gaming in children: A public health randomized controlled intervention study. Journal of Behavioral Addictions, 7(1), 52–61. https://doi.org/10.1556/2006.6.2017.087
Montag, C., Lachmann, B., Herrlich, M., & Zweig, K. (2019). Addictive features of social media/messenger platforms and freemium games against the background of psychological and economic theories. International Journal of Environmental Research and Public Health, 16(14), 2612. https://doi.org/10.3390/ijerph16142612
Q-2. El manuscrito sobresale en sus secciones de Método y Resultados. La metodología es robusta, con una explicación clara y completa del proceso de muestreo aleatorio multietápico y el uso de la técnica CART para el análisis de datos. El gran tamaño de la muestra y el enfoque riguroso para la selección de los participantes mejoran la fiabilidad del estudio. Además, la sección de Resultados está bien organizada y presenta los hallazgos de manera clara y concisa, lo que facilita el seguimiento de las relaciones entre las variables. El uso de técnicas estadísticas apropiadas refuerza aún más la validez de las conclusiones extraídas. Estas secciones contribuyen significativamente a la calidad general del manuscrito. El único aspecto que veo potencialmente mejorable es complementar las medidas de fiabilidad reportadas, como el Alfa de Cronbach, con el Omega de McDonald's. Dadas las limitaciones y suposiciones inherentes al uso del Alfa de Cronbach, la incorporación del Omega de McDonald's proporcionaría una estimación más matizada de la consistencia interna, especialmente en los casos en que los datos podrían no cumplir completamente con las suposiciones del Alfa. Esta adición reforzaría aún más la evaluación de la fiabilidad de los instrumentos utilizados en el estudio.
Response Q-2: 
We have recalculated the reliability indices using McDonald's Omega. We had not done this earlier because the statistical package used (SPSS – IBM, v. 24) does not provide this statistic. Therefore, to calculate it, we used the macro for SPSS by Hayes & Coutts (2020), which allows for the calculation of McDonald's Omega (see Section 2.2 Instruments, pages 4-5, Lines 183-209).
Video Game Experiences Questionnaire (CERV) (Chamarro et al., 2014). Instrument composed of 17 items, with a 4-point Likert response format (never/almost never, sometimes, quite often and almost always) that allows evaluating the problematic and abusive use of video games on any platform. It allows to obtain a total score and two subscales: evasion (8 items) and negative consequences (9 items). Likewise, cluster analyses offer a three-group solution, based on the following cut-off points: no problems with the use of video games (scores between 17 and 25 points); potential problems (between 26 and 38 points); and severe problems (between 39 and 68 points). Cronbach's alpha coefficients for the subscales are considered acceptable for the two subscales: Negative Consequences (α=.87) and Evasion (α=.86). In this study, Cronbach's alphas McDonald's Omega coefficient were .73 .729 and .82 .822 for each of the scales, respectively, and .89 for the global scale.
Emotional Repair Subscale of the Spanish version of the Trait Meta-Mood Scale (TMMS-24; Fernández-Berrocal et al., 2004). It is composed of 8 items that evaluate metaknowledge for the regulation of emotions, with a 5-point Likert-type response format (from 1 strongly disagree to 5 strongly agree). The psychometric properties show adequate internal consistency (α =.86) for the Emotional Repair subscale; in this study Cronbach's alpha McDonald's Omega coefficient was also adequate (α=.86) (ω=.86)..
Spanish version of the Strengths and Difficulties Questionnaire (SDQ) (Ortuño-Sierra et al., 2015). It is an instrument for clinical screening of mental disorders in childhood and adolescence, taking as a criterion the last 6 months. It is composed of 25 items, with a response format of three options (not true, a little true and absolutely true) grouped into 5 subscales (with 5 items each): Emotional Symptoms, Behavioral Problems, Hyperactivity, Relationship Problems with Peers and Prosocial Behavior. A difficulty score can also be obtained, which is the sum of the previous subscales, except Prosocial Behavior. The levels of reliability and validity for use in adolescents are adequate (Ortuño-Sierra et al., 2015). For this study, Cronbach's Alpha McDonald's Omega coefficient obtained was .70 712 for the Total Difficulties scale and .72 .723 for the prosocial behavior scale.
Spanish version of the Escala da Pasión (adapted to video games) (Chamarro et al., 2015). Passion is one of the elements of psychological processes present in various activities such as sports, leisure, work, interpersonal relationships or video games. This scale consists of 17 items, with seven answer options (strongly agree, fairly agree, agree, neither agree nor disagree, disagree, strongly disagree and strongly disagree). It allows us to obtain three dimensions: OP (6 items) and HP (6 items) and passion criteria (5 items). The internal consistency levels of the scale are adequate with a α=.81 for HP and α=.87 for the OP scale. The values of Cronbach's alpha statistic McDonald's Omega coefficient in the present study were suitable for both HP (α=.72) (ω=.73) and OP (α=.74) (ω =.75)..
Additionally, we made a brief mention in Section 2.5 Data Analysis (page 6, Lines 253-255).
To determine the levels of internal consistency of the instruments, we opted for McDonald's Omega coefficient, as it provides a more accurate estimate than Cronbach's alpha. To do this, we followed the procedure indicated by Hayes and Coutts (2020).
References:
Hayes, A. F., & Coutts, J. J. (2020). Use Omega rather than Cronbach's Alpha for estimating reliability. But…. Communication Methods and Measures, 14(3), 1–24. https://doi.org/10.1080/19312458.2020.1718629
Q-3. En la sección de Discusión, el manuscrito presenta una nueva exposición de los resultados comparándolos con los hallazgos de estudios previos. Sin embargo, esta comparación carece de interpretación y no proporciona posibles explicaciones para las similitudes o diferencias observadas. Por ejemplo, no hay una explicación sobre por qué los hombres exhiben tasas más altas de uso problemático de videojuegos o por qué la adicción a los videojuegos es más frecuente entre los adolescentes con TDAH. Por el contrario, la discusión en torno a la pasión está bien articulada y adecuadamente explicada. Incluir un debate más analítico de estas cuestiones, en particular de las variables que no se abordan adecuadamente, aumentaría la profundidad del análisis y ofrecería información valiosa sobre las implicaciones de los resultados.
Respuesta Q-3: 
In response, we have incorporated the following texts in Section 4. DISCUSSION (page 15, lines 577-594):
Moreover, in terms of content preference, video games that promote competition, aggression, and personal achievement tend to be more appealing to men. Additionally, men show a greater inclination toward action games and online multiplayer games, which tend to be more addictive due to their reward structures, intermittent reinforcements, and competitive environments (Männikkö et al., 2020).
…
This link can be explained by the difficulty individuals with ADHD have in regulating attention and impulses, as well as their tendency to seek rapid stimulation. In this regard, Nuyens et al. (2022) indicated that adolescents with ADHD tend to prefer video games that provide immediate gratification and high levels of activation—such as action and multiplayer games—because they offer quick reward cycles and constant reinforcements. Additionally, Boer et al. (2020) identified that problematic video game use in adolescents with ADHD may be influenced by the fact that video games provide a structured environment that these adolescents can control, unlike other areas of their lives that they often perceive as chaotic or difficult to manage. Video games provide a way to achieve success and control.
References:
Männikkö, N., Billieux, J., & Kääriäinen, M. (2020). Problematic gaming behavior in Finnish adolescents and young adults: Relation to game genres, gaming motives and self-awareness. Journal of Behavioral Addictions, 9(1), 256–266. https://doi.org/10.1556/2006.2020.00016
Nuyens, F., Deleuze, J., Maurage, P., & Billieux, J. (2022). Problematic gaming in adolescents with ADHD: The roles of impulsivity, game genres, and parental control. Journal of Behavioral Addictions, 11(1), 38–49. https://doi.org/10.1556/2006.2022.00004
Boer, M., Stevens, G., Finkenauer, C., & van den Eijnden, R. (2020). Attention Deficit Hyperactivity Disorder-Symptoms, Social Media Use Intensity, and Social Media Use Problems in Adolescents: Investigating Directionality. Child Development, 91(4), e853–e865. https://doi.org/10.1111/cdev.13334
References:
We have incorporated the following references in response to your requests:
Anjum, R., Nodi, N. H., Das, P. R., Roknuzzaman, A. S. M., Sarker, R., & Islam, M. R. (2024). Exploring the association between online gaming addiction and academic performance among the school-going adolescents in Bangladesh: A cross-sectional study. Health Science Reports, 7(9), e70043. https://doi.org/10.1002/hsr2.70043
Boer, M., van den Eijnden, R. J., Boniel-Nissim, M., Wong, S. L., Mathews, C., Morrell, H., & Molle, J. (2019). Video game addiction, ADHD symptomatology, and video game reinforcement. The American Journal of Drug and Alcohol Abuse, 45(1), 67–76. https://doi.org/10.1080/00952990.2018.1472269
Boer, M., Stevens, G., Finkenauer, C., & van den Eijnden, R. (2020). Attention Deficit Hyperactivity Disorder-Symptoms, Social Media Use Intensity, and Social Media Use Problems in Adolescents: Investigating Directionality. Child Development, 91(4), e853–e865. https://doi.org/10.1111/cdev.13334
Hayes, A. F., & Coutts, J. J. (2020). Use Omega rather than Cronbach's Alpha for estimating reliability. But…. Communication Methods and Measures, 14(3), 1–24. https://doi.org/10.1080/19312458.2020.1718629
Ip, E. J., Urbano, E. P. T., Jacobs, R. J., Lau, W. B., Clauson, K. A., Torn, R. A., Palisoc, A. J. L., & Barnett, M. J. (2021). The video gamer 500: Performance-enhancing drug use and Internet Gaming Disorder among adult video gamers. Computers in Human Behavior, 123, 106890. https://doi.org/10.1016/j.chb.021.106890
Krossbakken, E., Torsheim, T., Mentzoni, R. A., King, D. L., Bjorvatn, B., Lorvik, I. M., & Pallesen, S. (2018). The effectiveness of a parental guide for prevention of problematic video gaming in children: A public health randomized controlled intervention study. Journal of Behavioral Addictions, 7(1), 52–61. https://doi.org/10.1556/2006.6.2017.087
Männikkö, N., Billieux, J., & Kääriäinen, M. (2020). Problematic gaming behavior in Finnish adolescents and young adults: Relation to game genres, gaming motives and self-awareness. Journal of Behavioral Addictions, 9(1), 256–266. https://doi.org/10.1556/2006.2020.00016
Montag, C., Lachmann, B., Herrlich, M., & Zweig, K. (2019). Addictive features of social media/messenger platforms and freemium games against the background of psychological and economic theories. International Journal of Environmental Research and Public Health, 16(14), 2612. https://doi.org/10.3390/ijerph16142612
Nuyens, F., Deleuze, J., Maurage, P., & Billieux, J. (2022). Problematic gaming in adolescents with ADHD: The roles of impulsivity, game genres, and parental control. Journal of Behavioral Addictions, 11(1), 38–49. https://doi.org/10.1556/2006.2022.00004
Round 2
Reviewer 2 Report
Comments and Suggestions for Authors
I greatly appreciate your attention to all the suggestions and comments I made in the previous review. The manuscript has improved significantly, and I have no further observations.